Assessment of distinct effects of Parinari curatellifolia Planch.ex Benth Ethanolic leaf extract on glucose transport in different cell types

Omale Simeon 1 2 3
Aguiyi John C. 2
Ede Samuel 2
Ryalls Layla 3
Ye Runfei 3
Basbaydar Busra 4
Gould Gwyn W. gwyn.gould@strath.ac.uk 3
Bremner-Hart Shaun K. 3 4
1 Africa Centre of Excellence in Phytomedicine Research and Development and Department of Pharmacology and Toxicology, University of Jos , Jos , Nigeria
2 Department of Pharmacology and Toxicology, University of Jos , Jos , Nigeria
3 Strathclyde Institute of Pharmacy and Biomedical Sciences, University of Strathclyde , Glasgow , North Lanarkshire , United Kingdom
4 School of Molecular Biosciences, College of Medical, Veterinary and Life Sciences, University of Glasgow , Glasgow , United Kingdom
García-Contreras Rodolfo
Electronic publication date: 2025 Nov 10
Publication date: 2025
Volume: 13
Electronic Location ID: e20269
Received 2025 May 12; Accepted 2025 Sep 30
Copyright: ©2025 Omale et al.
Copyright year: 2025
Copyright holder: Omale et al.
License: This is an open access article distributed under the terms of the Creative Commons Attribution License, which permits unrestricted use, distribution, reproduction and adaptation in any medium and for any purpose provided that it is properly attributed. For attribution, the original author(s), title, publication source (PeerJ) and either DOI or URL of the article must be cited.
License URL: https://creativecommons.org/licenses/by/4.0/

Keywords: Diabetes, Natural products, Insulin, Glucose transport, Adipocyte, Small intestine, Caco-2 cells

Funding: The African Research Excellent Fund grant AREF-308-OMALE-F-C0818 Diabetes UK grant 18/0005847 This work was supported by a grant from the African Research Excellent Fund grant AREF-308-OMALE-F-C0818 (to Simeon Omale and Gwyn W. Gould) and Diabetes UK grant 18/0005847 (to Gwyn W. Gould). The funders had no role in study design, data collection and analysis, decision to publish, or preparation of the manuscript.

==============================
Extracts of Parinari curatellifolia Planch.ex Benth have been used as a traditional medicine in Sub-Saharan Africa for the management of various ailments including diabetes and has been shown to reduce plasma glucose levels in rat models of diabetes. Treatment of a range of mammalian cell lines with P. curatellifolia ethanolic leaf extract (PCE) for 24–48 h, typically between 0 and 100 µg/mL, revealed different actions: in 3T3-L1 adipocytes, PCE markedly inhibited insulin-stimulated glucose transport (50% inhibition at 100 µg/mL), whereas by contrast PCE-treatment of Caco-2 cells, a model of the intestinal epithelia at the same concentration, increased glucose transport ∼2-fold. This effect was accompanied by increased glucose transporter-1 (GLUT1) levels but is independent of changes in the level of Akt, Adenosine monophosphate-activated protein kinase (AMPK) or p38. Our data suggest that the antidiabetic effects of extracts of P. curatellifolia may arise by increased absorption of glucose from the gut and thus distribution to other cells/tissues. Our data further highlight the importance of screening metabolic actions of plant extracts against multiple cell lines, as these can often exhibit distinct cell-type-specific responses, and further suggest that relatively low doses of PCE (up to 100 µg/mL) could warrant investigation in in vivo models of disease.

Introduction

Diabetes is a major health care issue around the globe (Kahn, 2019; Saltiel, 2021). While current management approaches are efficacious, limitations exist. These include the high costs for developing nations and potential for confounding adverse drug events (Aschenbrenner, 2021; Gloyn & Drucker, 2018; Nair et al., 2018; Roglic & Norris, 2018; Xie et al., 2023). Consequently, research is increasingly examining the role of traditional therapies for anti-diabetic treatments (Omale et al., 2023; Usai, Majoni & Rwere, 2022).

More than 800 plant species have been suggested to have an anti-diabetic potential (Omale et al., 2023). For example, the roots of Zingiber officinale (ginger) lowered serum glucose levels in rat models of diabetes compared with the control animal models and has some efficacy in human studies (Al Syaad et al., 2019; Shidfar et al., 2015; Van et al., 2023; Zhu et al., 2018). Similar antidiabetic effects have been demonstrated with extracts of Momordica charantia (bitter lemon) on rat muscle cells (Shih et al., 2009; Tan et al., 2008). Traditional plant-based remedies are readily accessible to most remote and poor areas in Africa, making them of particular interest (Omale et al., 2023; Usai, Majoni & Rwere, 2022). However, it is worth noting that the translation of these kinds of therapies into effective pharmacological treatments is limited (see for example (Ndhlala, Moyo & Van Staden, 2010; Rahman et al., 2020; Talaulikar & Manyonda, 2011; Usai, Majoni & Rwere, 2022)), indicating that detailed mechanistic understanding of such extracts/plants is required.

Extracts of P. curatellifolia, notably from the leaves, have been used as a traditional medicine in Sub-Saharan Africa for the management of various ailments like cancer, diabetes, pneumonia, and inflammatory conditions (Gororo et al., 2016). Phytochemical screening of this plant indicated that it contained compounds including polyphenol, vitamin C and other secondary metabolites, such as alkaloid, anthraquinones, and glycosides which could contribute to its antioxidant properties (Gororo et al., 2016; Ogbonnia et al., 2011; Ogunbolude et al., 2011). Thus, this plant may exert an antidiabetic activity through its antioxidant properties as there is a strong link between oxidative stress and type-2 diabetes.

P. curatellifolia leaf extracts significantly reduced plasma glucose in rat models of diabetes and Ogbonnia et al. demonstrated a similar reduction in plasma glucose levels following treatment with P. curatellifolia ethanolic leaf extract (PCE) and root extract of Anthocleista vogelii in rat models of type 2 diabetes (T2D) (Ogunbolude et al., 2011). How such extracts modulate glucose transport into tissues remains to be evaluated.

Recent studies from our groups induced hyperglycaemia in Drosophila melanogaster by exposing flies to a high sugar diet to induce a type-2 diabetes-like state (Morris et al., 2012; Musselman et al., 2011). This hyperglyacemic state could be reversed by the classical antidiabetic agent, metformin, but also PCE in a dose-dependent manner (Omale et al., 2025). Characterisation of the leaf extract revealed that this effect may be a result of the presence of quercetin in these extracts (Omale et al., 2025). Quercetin has well-documented anti-diabetic actions (Basaldúa-Maciel et al., 2025; Bellavite, Fazio & Affuso, 2023; Hossein et al., 2024; Mantadaki et al., 2024). Such data suggested to us that mechanistic studies of this extract in mammalian cell culture models could offer clues to its mechanism of action.

We therefore examined the effect of PCE on glucose uptake in a range of mammalian cell systems. We report that PCE inhibits insulin-dependent glucose uptake in 3T3-L1 adipocytes independent of any effect on cell viability. By contrast, PCE increased glucose uptake by Caco-2 cells, a model of intestinal epithelia, with concomitant elevations in levels of glucose transporter-1 (GLUT1) protein. We speculated that a potential drug target for PCE may be AMP-activated protein kinase (AMPK), which is known to regulate glucose metabolism (Kahn et al., 2005; Rena, Hardie & Pearson, 2017; Steinberg & Hardie, 2023). We therefore compared growth rate and viability assays of wild type Schizosaccharomyces pombe or S. pombe lacking ssp2 (ssp2Δ); Ssp2p is the orthologue of AMPKα subunit. PCE had no effect on growth rate of either wild-type or ssp2Δ strains and exerted no effect on cell viability. Moreover, PCE did not reverse cell loss induced by the oxidative stress inducer H2O2. Consistent with this, we observed no effect of PCE on AMPK levels in mammalian cells. Further work is therefore required to define the mechanism of action of PCE in mammalian systems.

Materials and Methods

Plant extract

The leaf of P. curatellifolia Planch.ex Benth (wfo-0000817683) was collected from Oiji Ochekwu of Benue State, Nigeria in November 2020 and authenticated in the Department of Agric Extension Services, Federal College of Forestry Jos, Nigeria as detailed in Omale et al. (2025). The herbarium specimen was prepared and deposited in the Department of Pharmacognosy, Faculty of Pharmaceutical Sciences, University of Jos, Nigeria, and assigned the voucher specimen number UJ/PCG/HSP/11C26 (see Omale et al., 2025 for details). Extracts were prepared as outlined in Omale et al. (2025). In brief, leaves were air-dried at room temperature and pulverized to fine particles which were then stored at 4 °C until use. Eight hundred grams (800 g) of the powder was extracted using 70% ethanol (Sigma-Aldrich) in cold maceration for 72 h. The macerate was filtered using a Whatman filter paper (size 150 mm; Maidstone, England). The filtrate was freeze-dried, coded as PCE (P. curatellifolia extract) and stored at 4 °C until reconstitution for use in experiments.

Cell culture

Murine 3T3-L1 fibroblasts (RRID:CVCL_0123), rat H4-IIE hepatoma cells (RRID:CVCL_0284) and human Caco-2 cells RRID:CVCL_0025 were purchased from ATCC and grown and cultured as recommended by the suppliers and described in Harris et al. (1992) and Bremner et al. (2022) respectively.

2-deoxyglucose uptake

2-deoxyglucose uptake was assayed as outlined (Bremner et al., 2022; Roccisana et al., 2013). Cells were incubated in serum free Dulbecco’s Modified Eagle Medium (DMEM) for 2 h followed by Krebs-Ringer phosphate (KRP) (128 mM NaCl, 4.7 mM KCl, 5 mM NaH2PO4, 1.25 mM MgSO4, 1.25 mM CaCl2) for 20 min prior to stimulation with insulin. Assays were initiated by the addition of 50 µM 2-deoxy-D-glucose and 0.5 µCi 2-[3H]-deoxy- D -glucose (deGlc). Cells were incubated for 3-5 min (see legends to Figs. 1 and 2) followed by rapidly washing in ice-cold PBS. Cells were lysed with 1% (v/v) triton X-100 and radioactivity determined by liquid scintillation counting. In parallel, cells were treated with 10 µM cytochalasin B to determine non-specific association of 2-[3H]-deoxy-D-glucose.

Figure 1 P. curatellifolia effects on 3T3-L1 adipocytes.

(A) 3T3-L1 adipocytes at day-8 post differentiation were incubated with PCE at the indicated concentrations for 72 h. Thereafter, deGlc uptake was measured as described in cells left untreated or incubated with 100 nM insulin for 30 min. Shown are the absolute transport rates (pmol/million cells/minute) from triplicate biological replicates, each performed with three technical replicates per condition. (B) 3T3-L1 adipocytes treated as outlined in (A) were stained with Oli Red O. Representative images are shown; scale bars 100 µm. (C) Shows quantification of Oil Red O staining from triplicate biological replicates of the experiment shown in (B). In each case, four fields of cells were quantified per condition per biological replicate. PCE was without effect on Oil Red O staining at all concentrations tested. (D) Shows a cell viability assay for 3T3-L1 adipocytes incubated with PCE for 72 h at the concentrations shown. PCE was without effect on cell viability at all concentrations tested.

Figure 2 P. curatellifolia extract stimulated glucose uptake in Caco-2 cells.

(A) Caco-2 cells were incubated with PCE at the concentrations shown for 48 h and deGlc uptake assayed as described. Significant increases in deGlc uptake were observed at 50 µg/mL and 100 µg/mL PCE (*p = 0.026 and **p = 0.01, respectively). (B) Shows the result of an MTT assay for cells treated with PCE for 72 h at the concentrations indicated (*p < 0.01). (C) Shows representative anti-GLUT1 immunoblots of Caco-2 lysates prepared after either 24 or 48 h treatment with PCE at the concentrations shown. (D) Shows quantification of three biological replicates of the type shown in (C). Statistically significant elevations in GLUT1 levels are observed at 100 µg/mL PCE compared to control cells (*p = 0.02; **p = 0.018).

Oil Red O staining

Oil Red O staining was carried out as described in Bremner et al. (2022). Cells were fixed in 10% (v/v) formalin and washed briefly with 60% (v/v) isopropanol. Cells were stained with 5.14 mM Oil Red O in 60% (v/v) isopropanol for 10 min followed by four washes with water. Coverslips were dipped in Mayers Hematoxylin (Merck, Gillingham, UK) for 30 s followed by four washes in water prior to being mounted on glass microscope slides and photographed. For the quantification of Oil Red O stain, cells were incubated in 100% isopropanol for 15 min, the supernatant collected, and absorbance measured at 590 nm.

Cell lysate preparation and subcellular fractionation

Caco-2 cells were scraped into lysis buffer (50 mM Tris–HCl, pH 7.4 at 4 °C, 50 mM NaF, 1 mM Na4P2O7, 1 mM EDTA, 1 mM EGTA, 1% (v/v) Triton X-100, 250 mM mannitol, 1 mM DTT, Pierce™ Protease Inhibitor Tablet (Fisher Scientific, Loughborough, UK) and Phosphatase Inhibitor Cocktail Set II (Merck, Gillingham, UK)). Lysates were incubated on ice for 20 min before sedimentation at 21,910×g for 5 min at 4 °C. The supernatant was collected and stored at −20 °C (Bremner et al., 2022).

SDS-PAGE and immunoblotting

Sodium Dodecyl Sulfate-Polyacrylamide Gel Electrophoresis (SDS-PAGE) and immunoblotting was carried out as outlined (Bremner et al., 2022). Secondary antibody fluorescence was detected using the LI-COR Odyssey® SA system. Band intensity was quantified with ImageJ. Total protein was stained with Revert™ Total Protein Stain (Fisher Scientific, Loughborough, UK). Primary antibodies used are as follows; anti-panAKT (Cell Signaling Technology Cat# 2920, RRID:AB_1147620), anti-GLUT1 (Abcam Cat#ab115730, RRID:AB_10903230), anti-GLUT4 (A combination of rabbit polyclonal antibodies raised against the C terminus of GLUT4 and the N-terminus of GLUT4) as described, and anti-AMPK (Cell Signalling Technology Cat∼2532, RRID:AB_330331).

Yeast strains

Strains used were S. pombe AMPK wild-type (WT) (h- ade6-216 leu1-31 ura4-D18) and AMPK α-subunit orthologue ssp2Δ cells (h- ssp2::ura4+ ura4-D18 leu1-32) (Davie, Forte & Petersen, 2015). Basic yeast methods were as outlined (Moreno, Klar & Nurse, 1991). Strains were cultures in yeast extract (YE) media composed of 30 g/l of glucose, 5 g/l of yeast extract, 0.23 g/l of adenine, 0.23 g/l of uracil.

Yeast growth rate and viability analysis

18 h prior to experimentation, yeast culture suspensions were prepared and incubated overnight in a shaking incubator at 30 °C. The following day, suspensions were diluted to an OD600 = 0.1 and incubated with PCE as described in the legends. The OD600 was then recorded over time.

MTT assay

For cell viability assay, yeast cultures were diluted to OD600 0.6 with YE media in a 96-well plate. 10 µl of 10 mg/mL MTT (3-[4,5-dimethylthiazol-2-yl]-2,5 diphenyl tetrazolium bromide) was added to each well and incubated at 30 °C for 2 h. 100 µl of dimethyl sulfoxide (DMSO) was added to each well and the incubation continued for a further 1 h until formazan crystals dissolved and absorbance was measured at 570 nm using a microplate reader.

Glucose assays

For glucose oxidase assay, yeast strains were diluted to OD600 0.1 and were incubated with various concentrations of PCE (0, 10, 30, 100 and 300 µg/ml) for 2 or 18 h. One ml of cell suspension was sedimented by centrifugation at 3421 g for 3 min and the supernatant was collected and stored at −20 °C prior to assay. Each supernatant was diluted at 1:1000 and 50 µl assayed using a glucose oxidase-based method as outlined by the manufacturers (Sigma GAGO20).

Statistical analysis

All data were analysed in a GraphPad using one way analysis of variance (ANOVA) and 95% confidence level to test between the control and treated and in between groups, or unpaired t-tests.

Results

The utility of PCE as a potential anti-diabetic agent was first established using flies as a model system (Morris et al., 2012; Musselman et al., 2011; Omale et al., 2020). Studies have shown that high-sugar feeding leads to flies developing a state of ‘insulin resistance’, characterised by elevated levels of glucose and thus act as a model of T2D (Morris et al., 2012; Musselman et al., 2011; Omale et al., 2020). We showed that our regime of 10-day high sucrose diet supplementation resulted in a significant increase in glucose levels in the flies, and that incubation with increasing concentrations of PCE ameliorated this, consistent with an antidiabetic action of PCE (Omale et al., 2025). We reasoned this may arise via an activation of glucose transport in relevant tissues. Hence to further consider the mechanism of action of PCE, we turned to mammalian cell models.

Using 3T3-L1 adipocytes as a model of adipocytes, we observed that 2 h incubation with PCE (100 µg/mL) had no effect on either insulin-independent or insulin-stimulated glucose transport (Fig. S1). Because experiments in flies involved longer-term incubation with PCE, we repeated these studies using up to 72 h incubation with PCE. Under these conditions, a dose-dependent inhibition of insulin-stimulated deGlc uptake was consistently observed (Fig. 1A), but no effect on basal (unstimulated) rates was detected; maximal inhibition of insulin-stimulated glucose transport was observed at 100 µg/mL PCE, with higher concentrations not significantly increasing the extent of inhibition. In parallel studies, 72 h incubation of adipocytes with PCE did not exhibit changes in triglyceride content (as assessed by Oil Red O staining; Figs. 1B and 1C) which suggests no major effect on lipogenic pathways over these time frames and had no effect on cell viability (Fig. 1D). As the effects in flies showed an insulin-sensitizing effect of PCE, this effect on cultured adipocytes was not further investigated.

To explore potential effects in liver, we turned our attention to hepatoma cells to test the hypothesis that PCE may exert an effect on liver glucose metabolism. Using H4IIe cells, we consistently observed a reduction in insulin-independent glucose uptake in these cells, but this was accompanied by a significant effect on cell viability at comparable doses, with the PCE extract exhibiting toxicity event at relatively low doses (Fig. S2). Hence, this line was not further studied.

A key event in whole body glucose homeostasis is the absorption of glucose from the gut, a process also impaired in disease (Gromova, Fetissov & Gruzdkov, 2021; Gromova et al., 2022; Koepsell, 2020). We therefore examined glucose transport in Caco-2 cells as a model of the intestinal barrier (Harris et al., 1992; Morresi et al., 2022). Strikingly, we found that PCE consistently increased facilitative glucose transport in this cell type at concentrations of 50 µg/mL and above (Fig. 2A). This effect occurred concomitantly with increased cell growth, as reported by MTT assay (Fig. 2B, evident at 50 µg/mL, but not at higher concentrations). Treatment of cells with PCE for 24 or 48 h was found to result in an increased expression of GLUT1 (Figs. 2C and 2D).

We speculated that this effect may be mediated via increased expression of kinase molecules known to regulate glucose transport in this and other cells (Abbud et al., 2000; Barnes et al., 2002; Beg et al., 2017; Bergqvist et al., 2017; Bland et al., 2010; Boxer et al., 2006; Burcelin et al., 2003; Chung et al., 2021). However, we observed no change in the total levels of AMPK, Akt, JNK or Insulin Receptor β-subunit in response to 24 h or 48 h PCE treatment (Fig. 3).

Figure 3 P. curatellifolia extract does not change levels of signalling proteins in Caco-2 cells.

(A) Caco-2 cells were incubated with PCE at the concentrations shown for 48 h. Thereafter, cell lysates were prepared and 25 µg/lane probed using the indicated antibodies, using anti-actin as a loading control. Shown are representative immunoblots. (B) Shows quantification of three or more replicates of this type and revealed no significant changes in expression levels in response to PCE treatment.

We next considered that this effect may be mediated via an effect of PCE acting via pathways involving AMPK, as activation of AMPK is known to promote GLUT1 activity (Abbud et al., 2000; Barnes et al., 2002; Wu et al., 2013) (and also to impair insulin-stimulated glucose transport in 3T3-L1 adipocytes (Salt, Connell & Gould, 2000); cf. Fig. 1A). Our attempts to quantify AMPK activity using phospho-specific antibodies to this kinase did not provide consistent data, perhaps relating to the variable growth conditions observed in culture at any given time. However, this analysis is limited by using only a restricted set of time points of PCE treatment. We therefore decided to investigate using S. pombe which offers a genetically tractable experimental system which can be readily used to screen a range of conditions. AMP-activated protein kinase is encoded by the ssp 2 gene in S. pombe (Rena, Hardie & Pearson, 2017; Schonke et al., 2015; Steinberg & Hardie, 2023), we therefore compared cell growth and glucose utilisation in wild-type and ssp2Δ strains and found that over a wide range of concentrations, PCE extract did not affect either cell growth in solution (Fig. 4A) or using spot assays on solid media (Fig. S3). Similarly, glucose utilisation rates were unaffected in either strain (Fig. 4B). Oxidative stress has been demonstrated in many studies to participate in the progression of diabetes, including mediating an impairment of insulin action increased incidences of complications (Bhatti et al., 2022; Giacco & Brownlee, 2010; Veluthakal et al., 2024). Furthermore oxidative stress was reported to inhibit glucose uptake in Caco-2 (Morresi et al., 2022). Therefore, we considered whether PCE could protect S. pombe from oxidative stress induced by hydrogen peroxide in both wild-type and ssp2Δ strains. Although reduced cell viability was clearly observed in response to hydrogen peroxide in both strains, PCE exhibited no protective effects under these conditions (Fig. 4C).

Figure 4 P. curatellifolia extract does not change growth or glucose metabolism in S. pombe.

(A) Cell growth was assayed as described using either wildtype S. pombe or a strain lacking AMPK (ssp2Δ) in the presence or absence of PCE (300 µg/mL). No effect of PCE on growth was observed at this (or lower) concentrations (mean and S.E.M. for three biological replicates are shown). (B) In parallel experiments, we quantified the glucose levels in the culture media 2 h or 18 h (as indicated) after addition of PCE at the indicated concentration. No significant effects were observed; data is presented as mean of triplicate biological experiments (each performed with triplicate technical replicates) and normalised to control cells at time 0. (C) We quantified cell viability in yeast strains incubated with or without 0.1% (v/v) H2O2 as an inducer of oxidative stress. In both wild-type and ssp2Δ strains, oxidative stress treatment was accompanied by a loss of cell viability, but this was not affected by PCE in any of the groups.

Discussion

The growth of diabetes-related morbidity and mortality presents an increasing healthcare and economic burden across the globe. The need for novel therapies which can be widely exploited in under-developed economies has encouraged significant interest in plants and plant extracts that can drive metabolically relevant changes in insulin action (Akash et al., 2015; Mawire et al., 2021; Omale et al., 2023; Usai, Majoni & Rwere, 2022). This has been exemplified by a significant number of plant or plant extracts undergoing clinical trials (see Omale et al., 2023 for examples).

P. curatellifolia extracts are used as a traditional medicine in Sub-Saharan Africa for the management of diabetes (Gororo et al., 2016). Phytochemical screening of this plant identified antioxidant compounds including polyphenol, vitamin C and other secondary metabolites, which could contribute both to its antioxidant properties (Gororo et al., 2016; Ogbonnia et al., 2011; Ogunbolude et al., 2011) and potentially to its antidiabetic activity. Consistent with this, flavonoids from seeds of P. curatellifolia exhibit anti-hyperlipidemic and anti-atherogenic effects in rats (Crown et al., 2018) and seeds reduce plasma glucose levels in alloxan-induced diabetic rats (Ogbonnia et al., 2011). PCE can reverse a diabetic-like state in flies as effectively as metformin (Omale et al., 2025), and analysis of PCE revealed that a major component of the extract is quercetin (Omale et al., 2025), a molecule with known anti-diabetic actions (Ansari et al., 2022; Basaldúa-Maciel et al., 2025; Farhadi et al., 2024; Hossein et al., 2024; Mantadaki et al., 2024; Tylishchak et al., 2024). However, it remains to be established whether/if PCE exerts effects in mammalian cells and whether these effects are cell-type specific.

Thus prompted, here we considered the mechanistic basis of PCE action. Somewhat surprisingly, effects of PCE treatment of 3T3-L1 adipocytes revealed inhibitory effects of PCE on insulin-stimulated glucose transport, suggesting that the primary ability of PCE to reduce glucose levels in flies is unlikely to be via an effect on glucose transport in these cells (Fig. 1). Similarly, inhibition of glucose transport was observed in hepatoma cells, in this case accompanied by effects on cell growth and viability. By contrast, we observed a significant increase in glucose transport in Caco-2 cells, a model of the intestinal epithelial barrier (Fig. 2) This could suggest that an action of PCE may involve promoting glucose absorption from the gut and its more effective delivery to target tissues. We found that PCE treatment increased expression of GLUT1 in these cells (Fig. 2), concomitant with elevated glucose transport rates. Hence, our data suggest the hypothesis that this plant extract contains an active compound(s) that promote expression of the GLUT1 (or increased stability of the GLUT1 protein).

We were unable to demonstrate an effect of PCE-treatment on the expression of key signalling proteins in Caco-2 cells (specifically AMPK, Akt, p38 or the insulin receptor β-subunit; Fig. 3). Similarly, knockout of the ssp2 gene in yeast did not modulate any effect of the extract on yeast growth or glucose utilisation (Fig. 4). While indirect, these data suggest that modulation of AMPK activity is unlikely to be the driver of the effects observed on glucose transport and GLUT1 levels.

It is worthy of comment that both H4IIe cells and Caco-2 cells express GLUT1, yet the responses to PCE are clearly distinct between the cell lines. The mechanistic basis for this remains unclear, but this may be related to an apparent toxicity effect in Hepatoma cells which was not evident in Caco-2 cells. This differing response to PCE is also exemplified by our observation that insulin-stimulated glucose transport is inhibited by PCE, independently of effects on cell viability, in 3T3-L1 adipocytes. Thus, understanding the cell-specific responses to potential natural products should involve an analysis of multiple cells/tissues; generalising from a single cell line should therefore be avoided.

Conclusion

These studies reveal a tissue/cell line-specific effect of PCE on glucose transport and GLUT1 levels which may provide a mechanistic basis for the antidiabetic action of this plant.

It will also be important to consider effects of PCE on primary cells or in in vivo models to ascertain the validity of our data in more physiological systems, including its suitability to treat hyperglycaemia in a range of conditions. Similarly, comparative studies of leaf, shoot, and root extracts of this medicinally important plant may reveal interesting insight into other potentially important anti-diabetic molecules.

Supplemental Information

Supplemental Information 1 Acute addition of PCE does not modulate stimulated glucose transport

Using 3T3-L1 adipocytes as a model of adipocytes, we observed that 2h administration of PCE (100 µg/mL) had no effect on either insulin-independent or insulin-stimulated glucose transport. Shown is a representative experiment in which insulin was added at the concentration shown for 30 minutes after addition of PCE. Each bar is the average of triplicate technical replicates at each condition. Similar data was observed in three independent replicates of his type.

Supplemental Information 2 PCE treatment inhibits deGlc uptake and cell viability in H4IIE cells

(A) 24h treatment with the indicated concentration of PCE significantly inhibited basal (unstimulated) deGlc uptake in H4IIE cells. Shown is a representative experiment in which each bar is the average of triplicate technical replicates. (B) Shows MTT assay data from the same cell plating, in which a clear reduction in cell viability is evident. Similar data was observed in three independent replicates of his type.

Supplemental Information 3 WT S. pombe cell growth on solid YE media containing different concentrations of PCE

Wild type S. pombe cells were subjected to 10-fold serial dilution (1, 0.1, 0.01 and 0.001 OD600 measurement). Diluted cells were spotted onto PCE coated solid YE media plates at 0, 30, 300, & 3,000 µg/ml. Cell were grown at 30° C for 72 hours. Data from a representative experiment is shown, replicated three times.

Supplemental Information 4 Experimental data for Figure 3

Supplemental Information 5 Experimental data for Figure 4A, B

Supplemental Information 6 All other experimental datasets

Supplemental Information 7 Immunoblots

Abbreviations

AMPK AMP-activated protein kinase

DeGlc 2-deoxy-D-glucose

GLUT Glucose transporter

PCE ethanolic extract of Parinari curatellifolia leaves

T2D Type-2 diabetes

Additional Information and Declarations

Competing Interests

Author Contributions

Data Availability

Gwyn W. Gould is a Section Editor of PeerJ.

Simeon Omale conceived and designed the experiments, performed the experiments, analyzed the data, authored or reviewed drafts of the article, and approved the final draft.

John C. Aguiyi performed the experiments, analyzed the data, authored or reviewed drafts of the article, and approved the final draft.

Samuel Ede performed the experiments, analyzed the data, authored or reviewed drafts of the article, and approved the final draft.

Layla Ryalls performed the experiments, analyzed the data, prepared figures and/or tables, and approved the final draft.

Runfei Ye performed the experiments, analyzed the data, prepared figures and/or tables, and approved the final draft.

Busra Basbaydar performed the experiments, analyzed the data, prepared figures and/or tables, and approved the final draft.

Gwyn W. Gould conceived and designed the experiments, analyzed the data, prepared figures and/or tables, authored or reviewed drafts of the article, and approved the final draft.

Shaun K. Bremner-Hart conceived and designed the experiments, analyzed the data, authored or reviewed drafts of the article, and approved the final draft.

The following information was supplied regarding data availability:

The data is available in the Supplemental Files.

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
