# Peer review of "Assessment of distinct effects of Parinari curatellifolia Planch.ex Benth Ethanolic leaf extract on glucose transport in different cell types"

_PeerJ, doi:10.7717/peerj.20269_

## Round 0.1 · original submission · Major Revisions

· Academic Editor

Major Revisions

Please address all the concerns of the reviewers.

Reviewer 1 ·

Basic reporting

Title & Abstract

Title:
It may be helpful to mention in the title the type of extract utilized, i.e., ethanolic extract. This distinction is significant due to the potential differences in chemical composition and pharmacological effects among ethanolic, methanolic, and hydroalcoholic extracts. This recommendation is being made here considering different strengths and different alcohol or mixtures considered for study.
Example –
Assessment of distinct effects of Parinari curatellifolia Planch.ex Benth Ethanolic extract on glucose transport in different cell types
It is suggested that the manuscript would benefit from consistent use of professional English language to enhance clarity and avoid ambiguity.

Abstract:
Line 25: The distinction between the terms "adsorption" and "absorption" is scientifically significant and should be maintained throughout the manuscript. For example, line 237 of the discussion mentions "absorption," while the third-last line of the abstract, as well as other sections, use "adsorption." Since the current research explores the transfer of glucose molecules across potentially multiple interfaces, such as the mucosal epithelium and muscularis mucosa, the term "absorption" may be more appropriate unless a single interface interaction is definitively established.
Lines 18-19: The phrase "reverse glucose level" used in the abstract may not accurately reflect the scientific understanding of diabetes models. A more precise approach would be to reference the regulation of hyperglycemia, management of insulin resistance, and maintenance of euglycemia, supported by biochemical parameters such as insulin levels and HOMA-IR.
Line 20: In the abstract, certain key elements require clarification. For example, the abbreviation "PCE" should be defined. Additionally, it would be beneficial to briefly mention the number of experiments, dose ranges tested, statistically significant findings, and the overall conclusion derived from these findings. The method section of the abstract should be self-explanatory, ideally including the interventions, cell lines used, dosage levels, and p-values.
Line 27: The concluding lines of the abstract might be rephrased to more precisely reflect the purpose and outcome of the research. For instance, it could be highlighted that a 50 µg/mL in vitro dose of PCE demonstrated potential and warrants further investigation in in vivo models.

Introduction
Lines 41-42: The introduction and background sections would benefit from a broader context. While the authors mention the high cost of diabetic medications as a limiting factor in developing countries (lines 41-42), it is respectfully noted that additional global challenges, such as adverse drug events and the risk of resistance, are also relevant. For instance, some antidiabetics have been withdrawn due to risks such as bladder carcinoma. Including these broader issues would enhance the introductory rationale.
Line 55: Given the global readership of the target journal, the use of layman’s terminology should be minimized. For instance, line 55's phrasing, "its sticks can be chewed" and "drinks can be made," could be replaced with scientific descriptions such as "the extract from the stem can be administered orally or formulated in a hydroalcoholic solution."
Antioxidant activity is shown by almost every existing plant species. However, the clinical pharmacology translation has happened very rarely.
Ndhlala AR, Moyo M, Van Staden J. Natural Antioxidants: Fascinating or Mythical Biomolecules? Molecules. 2010; 15(10):6905-6930. https://doi.org/10.3390/molecules15106905
Talaulikar VS, Manyonda IT. Vitamin C as an antioxidant supplement in women's health: a myth in need of urgent burial. Eur J Obstet Gynecol Reprod Biol. 2011 Jul;157(1):10-3. doi: 10.1016/j.ejogrb.2011.03.017. Epub 2011 Apr 20. PMID: 21507551.
You may use this argument as well to make up a flow of discussion in the introduction section.
The introduction section avoids referencing quercetin, a previously identified phytochemical in Parinari curatellifolia Planch. ex Benth. It is recommended that this compound and its relevance be acknowledged early on to strengthen the scientific foundation.
https://doi.org/10.1016/j.prenap.2025.100231

Lines 65-66: could be interpreted as suggesting beta-cell preservation or reversal of pancreatic necrosis. If this is the case, the methodology, including the induction model (e.g., alloxan or streptozotocin), should be briefly described.
Line 62: Discussion of LDL levels appears tangential and may be omitted unless directly relevant to the hypothesis.
Lines 75-76: The hypothesis presented appears inconsistent. If PCE inhibits insulin-dependent glucose uptake in adipocytes, its classification as antidiabetic should be reconsidered or more thoroughly justified, especially since insulin-independent uptake is limited to tissues such as the brain and red blood cells.
Lines 81-83: The transition between experimental foci disrupts the flow of the narrative. A more structured progression would improve readability.
While the manuscript is well-referenced, it is suggested that the number of self-citations be reduced to maintain objectivity and broaden the manuscript's relevance. Example - Omale et al, 2023, 2025 throughout the manuscript.

Figures & Tables
There are no tables but 4 figures in the manuscript. Figures are of good quality, readable, and clear. Figures 1,2, and 4 require modifications -
Figure 1:
• Figure 1A includes the 200 µg/mL dose, which is not mentioned in Figures 1B–1D. Similarly, the 10 µg/mL dose appears in Figures 1B–1D but not in 1A. Consistency is recommended in the revision of figures as well as the text part of the results section.
• The relevance of Oil Red O staining in this context may be limited, as lipid accumulation does not directly relate to the research question focused on antidiabetic activity.
• Cell viability data presented in Figure 1D (approximately 65% at 10 µg/mL) suggest cytotoxicity, which is not discussed in the results or discussion sections. This warrants attention.
Figure 2:
• The Y-axis in Figure 2B differs from that in Figure 1D. Standardizing the axis scales is recommended.
• Cell viability appears inconsistent, with low viability at both 10 and 100 µg/mL and a higher value at 50 µg/mL. These trends should be addressed explicitly in the results and discussion sections.
Figure 3 appears acceptable in format and clarity.
Figure 4:
• In Figure 4B, the X-axis legends are repeated in the upper and lower panels. Clarification is needed to indicate whether these represent different time points (e.g., 24h and 48h).

Experimental design

Materials and Methods

Replication of the experiment is difficult considering some important issues raised herewith. The time frame of the study is not mentioned in the research paper. In vitro experiments are generally short experiments. The outcomes are assessed within a few days. Authors have mentioned statistical test p-value details in the figure legends. However, it can be better placed in the statistical analysis section just before the result section. There are a few concerns pertaining to the material and methods section.
The authors have undertaken a series of in vitro experiments across different cell lines (Murine 3T3-L1 fibroblasts, rat H4-IIE hepatoma cells, human Caco-2 cells, and yeast strains), which is commendable. Ethical approval is not mandatory for in vitro studies.
However, certain methodological issues require clarification:
• Line 89: The plant leaves were collected five years ago (Year 2020 mentioned on Line 90). Information regarding storage conditions and measures taken to preserve biological activity would be valuable.
• Line 97: The location "Maidstone, England" could be placed in parentheses.
• Line 103: Media abbreviations such as DMEM should be expanded upon first use.
• Line 104: A detailed list of buffer constituents is not necessary for a scientific background audience.
• Line 110: Oil Red O staining does not appear central to the antidiabetic hypothesis and could be omitted.

Validity of the findings

Results

Line 156 onwards, Discussion paragraphs are put in the results section of the manuscript. Please reorganize, considering so many references in the result section. It is a space provided to showcase the author's observations and for the compilation of other experiments happened in the past. Authors are requested to see the comments for figures 1-4. The results, which should have been mentioned in the text section, are mentioned in the figure legends.
Line 156 onwards, throughout the result section - The current structure of the manuscript does not conform to Journal standards. Specifically, discussion content appears in the results section, while figure legends include analytical conclusions. The manuscript's conclusion appears to rely heavily on prior work rather than the current findings.
All figures are of good quality and appropriately labelled; however, the legends could be more concise. It is suggested that detailed statistical notations be streamlined for better readability.
Raw data were not supplied. The alignment and presentation of experimental results should be refined for clarity.
Lines 156-163. The results section should begin with current experimental findings rather than prior observations. The use of the word “administration” (line 166) could be revised to “exposure” or “contact time” for greater precision.
Lines 185-186: Claims of dose-dependent activity are not substantiated by the data in Figure 2A, which lacks clear dose-response trends.
Lines 192-213: References to previous publications could be reduced to better highlight the novel contributions of the current study.
Line 188 specifies dosing information in the results section, despite the use of a broad dose range (10 to 300 µg/mL). Including this information more comprehensively would enhance clarity.

Discussion

The research question addressed is original and relevant within the journal’s scope. The investigation into the antidiabetic potential of Parinari curatellifolia is timely and significant given current pharmacological developments.
The manuscript describes a bit of haphazardly addressed research question aiming to elucidate the mechanism of action of the plant extract. The relevance and novelty of the work are clearly absent.
The rationale for the study is derived from previous work by the authors on Drosophila melanogaster, which suggested possible antihyperglycemic activity. This provides a meaningful knowledge gap for the current investigation.
The discussion summarizes some of the findings; however, some enhancements are recommended:
• Begin with a clear statement of the study's objectives and importance.
• Organize subsequent paragraphs by major findings, comparing them to previous literature and noting any similarities or discrepancies.
• Conclude each paragraph with a statement on the study’s novelty.
Lines 255–258: The final conclusion should be based solely on the current study’s results. It is not advisable to draw conclusions from prior research unless directly linked to current findings.
The discussion would benefit from greater attention to:
A. Novelty – Highlight how this study extends previous findings, including the prior identification of quercetin in the extract.
B. Impact – Address how these results contribute to pharmacology and diabetes research. Potential use in type 2, type 1 diabetes, as a single therapy or as an add-on therapy, etc.
C. Limitations – Discuss the scope of the experiments and any alternate mechanisms (e.g., GLP-1) or durations that were not evaluated.
D. Future Directions – Suggest next steps such as in vivo validation, dose optimization, and progression toward early-phase clinical trials.
For replicability, it is important to contextualize observed effects, such as cytotoxicity, with findings from existing literature.
There are just 5 references used in the discussion section, and out of those 5, two are self-citations. Authors are encouraged herewith to use more in vitro and in vivo references.
e.g.
Rehman G, Hamayun M, Iqbal A, et al. In Vitro Antidiabetic Effects and Antioxidant Potential of Cassia nemophila Pods. Biomed Res Int. 2018;2018:1824790. Published 2018 Jan 23. doi:10.1155/2018/1824790
Kifle ZD, Yesuf JS, Atnafie SA. Evaluation of in vitro and in vivo Anti-Diabetic, Anti-Hyperlipidemic and Anti-Oxidant Activity of Flower Crude Extract and Solvent Fractions of Hagenia Abyssinica (Rosaceae). J Exp Pharmacol. 2020;12:151-167. Published 2020 Jun 9. doi:10.2147/JEP.S249964
Annapandian VM, Sundaram RS. In vitro Antidiabetic Activity of Polar and Nonpolar Solvent Extracts from Leucas aspera (Willd.) Link Leaves. Pharmacognosy Res. 2017;9(3):261-265. doi:10.4103/pr.pr_141_16

Conclusion

The ending lines of discussion, i.e, Line 255 to 258, discusses previous paper findings (Omale 2025). It is a space for talking about the conclusion, limitations, and future direction.

Reviewer 2 ·

Basic reporting

1. Although the authors put the "PCE" in the abbreviations section, the reviewer strongly suggests that the authors should indicate that the full name and its abbreviation are given upon first mention in the abstract. In the abstract, the reader can't understand what PCE is since there is no explanation.

2. The scientific name "Parinari curatellifolia" is written in full twice in the abstract. The second instance should be abbreviated as P. curatellifolia, following standard taxonomic conventions. The same applies throughout the main text. If the author would like to stick to using the abbreviation "PCE" after your first mention, that's okay, too. Please ensure consistency throughout the manuscript.

3. Page 8, line 107 / Page 10, line 139 - The reference to figure legends is vague. Figure legends are referenced without indicating which figure. Please clarify.

4. The yeast strain should be referred to using its full scientific name for the first mention in the manuscript: Schizosaccharomyces pombe.

5. Page 11, line 173: "Figure 1B,C" should be corrected to "Figures 1B and 1C." Similar formatting issues occur elsewhere and should be systematically corrected throughout the manuscript.

6. Page 8, line 156: The common name “flies” should be included when referring to Drosophila to enhance clarity.

7. In some figures, the abbreviation "DeGlc" should use a lowercase “d” for “deoxy”.

8. In some figure legends, the cell line name “CACO-2” should be corrected to “Caco-2.”

9. Figure axis labeling should be standardized. For example, Figure 1A should follow the same format as Figure 1C, including consistent labeling of the x-axis with “PCE (μg/mL).” Units within individual bar plots can be removed once axis units are standardized.

10. Figures 1C and 1D lack appropriate y-axis labeling or units. Please revise accordingly.

11. In the legend for Figure 1, use the abbreviation “deGlc” instead of the full name “2-deoxy-D-glucose” since the authors already introduced the abbreviation upon the first mention.

12. Figure 1D y-axis should follow the formatting used in Figure 2B for consistency.

Experimental design

1. For the cell culture conditions, please indicate the details about the recommendations of the cell line suppliers. If a specific protocol or reference is followed, it should be cited.

2. For the supplementary figures and Figure 4, the reviewer doesn't see that these experiments were conducted in triplicate. Please indicate whether the data are based on triplicate measurements, and include standard deviation (SD) error bars if applicable.

Validity of the findings

1. The study used only the leaf part of the plant. Please clarify the rationale behind this choice. Is the leaf the primary part used in traditional treatments, or are other parts of the plant also traditionally employed? If other plant parts are used for the treatment, a comparative phytochemical analysis between each part, such as by using LC-MS, would strengthen the study. Also, it should be clearly stated that the leaves of P. curatellifolia are used in the title.

2. The Discussion section lacks sufficient detail about the compounds previously identified from P. curatellifolia. Please expand this section to discuss known phytochemicals, especially secondary metabolites, their reported bioactivities, and how they may relate to the current findings. In addition, as the reviewer mentioned in the previous comment, it is strongly encouraged that LC-MS-based tentative identification of the extract’s constituents should be included, accompanied by chromatograms and a compound identification table.

---

## Round 0.2 · Minor Revisions

· Academic Editor

Minor Revisions

Please address the minor comments by one reviewer and this comment to me:

"Potential antidiabetic activity of PCE has been reported by previous studies, and the molecular target was not identified by the present investigation. Thus, this work is not valuable for a potential application. The author may think about insulin receptor and PI3K and their related proteins for a potential antidiabetic target of P. curatellifolia in their future research work."

Reviewer 2 ·

Basic reporting

Most of the suggestions have been appropriately addressed.

However, here are additional minor comments authors need to address.

1. Figure 2A, DeGlc -> deGlc
2. Line 244, flavenoids -> flavonoids
3. The reviewer is uncomfortable seeing "novel" in lines 392 and 398. Authors should be cautious when using the expression "novel". The reviewer suggests either replacing it with something like 'previously unreported' or deleting it.

After revising these, it is suitable for publication.

Experimental design

no comment

Validity of the findings

no comment

Reviewer 3 ·

Basic reporting

Journal: PeerJ
Manuscript ID: ID-118179
Manuscript Type: Review
Title: Assessment of distinct effects of Parinari curatellifolia Planch.ex Benth Ethanolic leaf extract on glucose transport in different cell types
Authors: Simeon Omale, John C. Aguiyi, Samuel O. Ede, Layla Ryalls, Runfei Ye, Busra Basbaydar, Gwyn W. Gould, Shaun K. Bremner-Hart

In the present investigation, extract of the leaves of Parinari curatellifolia Planch.ex Benth (PCE) has been evaluated for its potential antidiabetic activity. In 3T3-L1 adipocytes, PCE inhibited insulin-stimulated glucose transport while, in Caco-2 cells, PCE increased glucose transport. These activities correlated with increased glucose transporter-1 (GLUT1) levels but were independent of changes in the level of Akt, adenosine monophosphate-activated protein kinase (AMPK) or p38. These data indicate that PCE could show antidiabetic activity, which may result from increased absorption of glucose from the gut. Potential antidiabetic activity of PCE has been reported by previous studies, and the molecular target was not identified by the present investigation. Thus, this work is not valuable for a potential application. The author may think about insulin receptor and PI3K and their related proteins for a potential antidiabetic target of P. curatellifolia in their future research work.

Experimental design

Not good.

Validity of the findings

Not good.

Additional comments

The data reported are not supportive of any publications.

---

## Round 0.3 · accepted · Accept

· Academic Editor

Accept

Thanks for addressing all comments.

Reviewer 2 ·

Basic reporting

no comment

Experimental design

no comment

Validity of the findings

no comment